# A Review on Ultrasonic Neuromodulation of the Peripheral Nervous System: Enhanced or Suppressed Activities?

**Bin Feng \***, **Longtu Chen and Sheikh J. Ilham**

Department of Biomedical Engineering, University of Connecticut, Storrs, CT 06269, USA;
longtu.chen@uconn.edu (L.C.); sheikh.ilham@uconn.edu (S.J.I.)
**\*** Correspondence: fengb@uconn.edu; Tel.: +1-860-486-6435

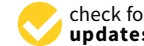

**Featured Application:** **The current review summarizes our recent knowledge of ultrasonic neuromodulation of peripheral nerve endings, axons, and somata in the dorsal root ganglion. This review indicates that focused ultrasound application at intermediate intensity can be a non-thermal and reversible neuromodulatory means for targeting the peripheral nervous system to manage neurological disorders.**

**Abstract:** Ultrasonic (US) neuromodulation has emerged as a promising therapeutic means by delivering focused energy deep into the nervous tissue. Low-intensity ultrasound (US) directly activates and/or inhibits neurons in the central nervous system (CNS). US neuromodulation of the peripheral nervous system (PNS) is less developed and rarely used clinically. The literature on the neuromodulatory effects of US on the PNS is controversial, with some studies documenting enhanced neural activities, some showing suppressed activities, and others reporting mixed effects. US, with different ranges of intensity and strength, is likely to generate distinct physical effects in the stimulated neuronal tissues, which underlies different experimental outcomes in the literature. In this review, we summarize all the major reports that document the effects of US on peripheral nerve endings, axons, and/or somata in the dorsal root ganglion. In particular, we thoroughly discuss the potential impacts of the following key parameters on the study outcomes of PNS neuromodulation by US: frequency, pulse repetition frequency, duty cycle, intensity, metrics for peripheral neural activities, and type of biological preparations used in the studies. Potential mechanisms of peripheral US neuromodulation are summarized to provide a plausible interpretation of the seemly contradictory effects of enhanced and suppressed neural activities of US neuromodulation.

**Keywords:** ultrasound; neuromodulation; single-unit recording; pain; sciatic nerve; compound action potential

---

## 1. Introduction

Our nervous system consists of the central and peripheral nervous systems (CNS, PNS), which have similar ion channel/modulator compositions [1,2]. In the CNS, functional neural circuits implicated in different neurological diseases overlap significantly with one another (e.g., overlapping circuits for dementia, aphasia, and Alzheimer's disease [3]), and certain neural circuits are not restricted to one region but spread throughout the brain (e.g., the pain matrix [4]). Collectively, this has led to the challenge of developing selective and effective drugs that target certain neurological diseases with limited off-target side effects. The recent advancement of optogenetic neuromodulation offers the much-needed selectivity at the expense of invasive and permanent gene modification of neural tissues.

On the other hand, neuromodulation with focal delivery of physical energy to an affected area in patients has drawn growing attention as a non-drug alternative for managing neurological diseases and symptoms.

The most widely used stimulus modality in neuromodulation is electrical stimulation, including electrical stimulation of the brain for treating movement disorders, stroke, tinnitus, depression, and addiction [5], as well as stimulation of the spinal cord and peripheral nerves for managing various types of chronic injuries and pain [6–8]. Besides electrical stimulation, other modalities of stimuli have been implemented that allow no-contact delivery of physical energy deep into the neuronal tissue, including focused ultrasound [9], transcranial electromagnetic stimulation (TMS) [10,11], and infrared light pulses [12]. TMS uses a strong magnetic field of 2 to 3 T to evoke current pulses in the tissue, but it faces the challenge of precisely localizing the activated areas in the brain because of the electrical and magnetic anisotropy of the brain and skull tissues. Infrared light pulses affect neural activity by delivering a spatially precise thermal stimulus, but the local heating of the targeted region remains a concern of the method. On the other hand, ultrasound (US), as a mechanical wave operating at 250 KHz to 50 MHz, allows spatially and temporally precise delivery of energy deep into the tissue with controllable heating. Hence, US can be considered as an ideal means for non- or minimally invasive neuromodulation.

The effects of US on the CNS have been shown to disrupt the blood brain barrier and evoke excitatory and/or inhibitory responses in both motor and sensory neurons [13–23]. In addition, transcranial ultrasonic stimulation (TUS) as a non-invasive brain stimulation technique has been investigated in primates. TUS of the frontal eye field evokes a transient increase of neural activities in the supplementary eye field in awake macaques as measured by multi-channel single-unit recordings [24]. In addition, a 40 s-long TUS of certain brain regions can lead to long-lasting neuromodulatory effects up to two hours post-stimulation, according to a functional MRI study in macaques [25]. Especially, TUS of the supplementary motor area and frontal polar cortex causes those brain regions to interact more selectively with the rest of the brain. Outcomes of those researches have led to the successful translation of US stimulation, after approval by the U.S. Food and Drug Administration (FDA), to treat medical-refractory patients with essential tremor [26]. Electrophysiological recording from the rat hippocampal dentate gyrus has been reported to simultaneously enhance (at the fiber volley) and suppress (in dendritic layers) the compound action potentials (CAP) in response to US stimulation [27]. In line with this, another ex vivo study based on calcium imaging, shows low-intensity focused pulsed US can evoke electrical activities in the mouse hippocampal slices [28]. However, two most recent studies suggest that US neuromodulation of the CNS does not directly activate brain regions in mice, but through an indirect auditory cochlear pathway [29,30]. Nonetheless, these recent results from whole-organ and whole-animal studies do not invalidate the prior studies in reduced systems, which show apparent neuromodulation by focused US in the absence of a functioning auditory system, like in *Caenorhabditis elegans*, tissue culture, retina [31], and brain slices (reviewed in reference [9]).

Peripheral neuromodulation targets the PNS to preclude off-target CNS effects and thus is even more selective than the CNS neuromodulation. PNS neuromodulation is particularly appealing to treat chronic pain, as the pain circuitry in the CNS is complex and widespread. State-of-the-art peripheral neuromodulatory strategies to treat chronic pain include spinal cord stimulation that targets peripheral nerve entry to the spinal cord [8,32], peripheral nerve field stimulation that targets a region of tissue [33,34], direct peripheral nerve stimulation [35], and the recent dorsal root ganglion (DRG) stimulation of sensory neural somata [36,37]. US neuromodulation of the PNS is far less advanced compared to CNS neuromodulation (see reference [38] for a review), and its mechanisms of action remain unclear. In this paper, we comprehensively review the methods and outcomes of studies on US effects in altering neural activities in the peripheral nerve axons, endings, and DRG. The seemly contradictory effects of US on peripheral nerves are discussed in the context of the different study designs and methods. In addition, we summarize the existing theories that account for the effects of US neuromodulation on peripheral nerves and DRG.

## 2. Peripheral Ultrasonic Neuromodulation—Technical Specifications

US waves are acoustic waves caused by mechanical vibrations at frequencies above 20 kHz (the upper hearing range of the human ear). The frequency ($f_0$) of the mechanical vibration source determines the frequency of the propagating US wave. US waves propagate through both the liquid and solid media, in the form of vibrating media particles, according to the following governing equation:

$$\nabla^2 p - (\rho\kappa)\frac{\partial^2 p}{dt^2} = 0 \tag{1}$$

in which $p$ is the media pressure, $\rho$ is the media density, and $\kappa$ is the media compressibility. The US wave speed $c$ is equal to $\frac{1}{\sqrt{\rho\kappa}}$. For a simple monochromatic vibration source, the pressure of the US wave derived from Equation (1) takes the form of a harmonic plane wave:

$$p(r,t) = A\cos(2\pi f_0 - kr) \tag{2}$$

in which $r$ is the distance vector from the source, $A$ is the amplitude of the wave, and $k$ is the wave number equal to $2\pi/\lambda$.

Our current understanding of PNS neuromodulation is limited by the controversial outcomes of different experimental studies, which are most likely affected by several critical parameters relevant to the US itself, as well as by the metrics used to evaluate US neuromodulation. Accordingly, the following parameters will be discussed in detail in this review: US frequency, pulse repetition frequency, duty cycle, intensity, metrics for peripheral neural activities, and type of biological preparations.

### 2.1. US Frequency

The frequency of the US wave is determined by the central frequency of the acoustic vibration source, generally, a US transducer. The US frequency used in the biomedical field ranges from 0.25 to 50 MHz. The intensity of a US beam attenuates exponentially with the propagation distance because of both absorption and scattering processes. Bones, especially cancellous bones, cause more severe US scattering than soft tissues due to their material heterogeneity. Also, the US absorption coefficient is much higher in bones than in the soft tissues. In addition, higher-frequency components are prone to relatively rapid attenuation. Hence, to penetrate the bony skull and the skin, non-invasive US neuromodulation generally operates at a lower frequency range from 0.5 to 3 MHz [38]. In contrast, high-frequency US (>3 MHz) is used in invasive surgeries to ablate tissues by implementing its high absorption coefficients to heat the local tissues [39]. In addition, high-frequency US at low intensity is widely used in US imaging to enhance resolution via reduced wavelength in the sub-millimeter range. One exception for the use of high-frequency US in neuromodulation (up to 43 MHz) is the activation of the retina with high spatial resolution for vision restoration [40], in which the penetration of the skull is not required.

### 2.2. Pulse Repetition Frequency

US neuromodulation usually does not use continuous waves but burst of waves with certain pulse-width, as shown in Figure 1. The pulse repetition frequency reflects the frequency of the burst, which can be orders of magnitude lower than the US frequency. The pulse width can be as short as a few US cycles and as long as the pulse repetition period. The pulse repetition frequency adds an additional frequency component to the spectrum of the US wave and can be critical in activating auditory nerve endings and other low-threshold mechanoreceptors.

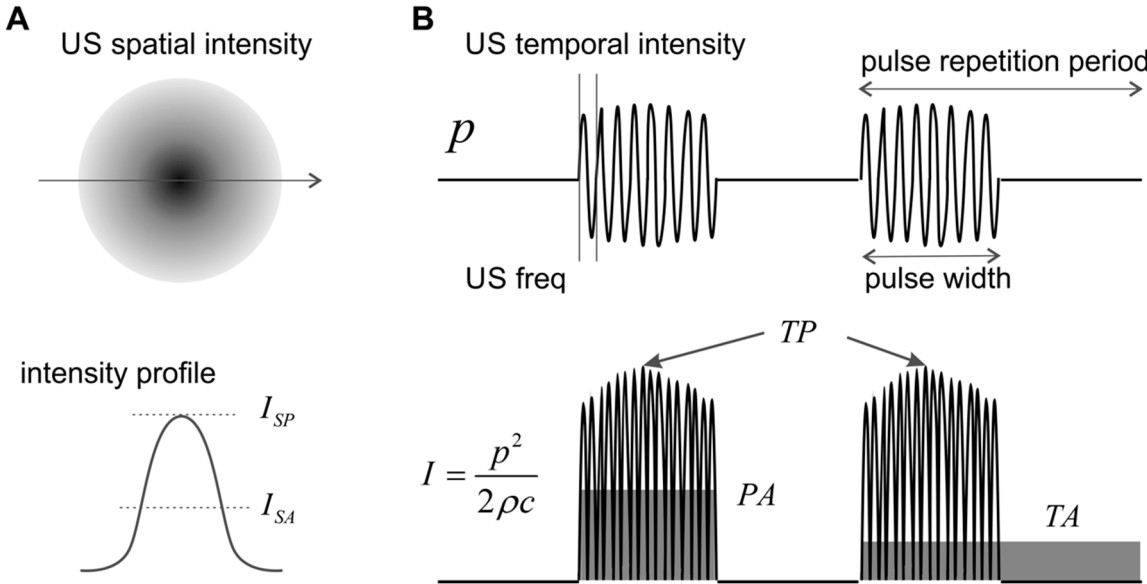

**Figure 1.** Schematics of spatial and temporal ultrasound (US) intensity. (**A**) The spatial intensity distribution peaks at the focal location and attenuates quickly outside the focus. (**B**) Three different temporal averages of US intensity. *p*: pressure; *I*: intensity; SP: spatial peak; SA: spatial average; TP: temporal peak; PA: pulse average; TA: temporal average.

## 2.3. Duty Cycle

The duty cycle is defined as the ratio between the pulse width and the pulse repetition period, which is generally less than 2% for diagnostic US imaging devices [41] but can be as high as 100% in US neuromodulation.

## 2.4. Intensity

The strength of an ultrasonic wave is characterized by its intensity, usually in the unit of watts per centimeter square (W/cm$^2$), i.e., the average power per unit cross-sectional area evaluated over a surface perpendicular to the propagation direction. For acoustic plane waves, the intensity is related to the pressure amplitude by:

$$I\left(\mathrm{W/cm}^2\right) = \frac{p^2}{2}\sqrt{\frac{\kappa}{\rho}} = \frac{p^2}{2\rho c}$$

As shown in Figure 1A, the spatial distribution of US intensity peaks at the focal location and attenuates quickly outside the focus. To evaluate US intensity in neuromodulation, spatial peak (SP) intensity is more widely used than spatial average (SA) intensity. In the time domain, the instantaneous intensity can be calculated from the pressure plot, as shown in Figure 1B. The temporal peak (TP) intensity is the maximum US intensity, the pulse average (PA) intensity is the average intensity within the pulse width, and the temporal average (TA) is the average intensity for several pulse repetition cycles. In experimental and clinical studies, US intensity is generally quantified as spatial peak-temporal peak (SPTP), spatial peak-pulse average (SPPA), and spatial peak-temporal average (SPTA) intensities. In order to compare studies, it is worth emphasizing that, for the same US wave, the magnitude of $I_{SPTP}$, $I_{SPPA}$, and $I_{SPTA}$ is in descending order, and $I_{SPTP}$ can be orders of magnitude higher than $I_{SPTA}$ for US pulses with low duty cycle. Physically, $I_{SPTP}$ reflects the highest spatial intensity in the US beam and is closely related to potential mechanical effects and cavitation in the tissue. $I_{SPTA}$ measures the highest spatial intensity averaged over the pulse repetition period and is related to the magnitude of the thermal effect. US neuromodulation generally operates at low-intensity levels to avoid cavitation and is more concerned with the local thermal effect. Thus, $I_{SPTA}$ appears to

be a more suitable intensity indicator for peripheral neuromodulation studies to avoid any prominent thermal effectors.

US stimulation of $I_{SPTA}$ below 1 W/cm$^2$ is generally considered as low-intensity, and the FDA has approved the application of US in patients with a maximum $I_{SPTA}$ of 0.72 W/cm$^2$ for diagnostic purposes, which is, presumably, safe enough also for therapeutics [42]. Several studies have reported neuromodulatory effects on the CNS, with US intensity <1 W/cm$^2$ [15,20–22,28]. So far, there seem to be no studies reporting appreciable neuromodulatory effects on the PNS with US intensity <1 W/cm$^2$ [43]. On the other hand, $I_{SPTA}$ over 200 W/cm$^2$ is generally considered as high-intensity US and has been tested in a number of clinical trials [44,45], after being approved by the FDA, for the ablation of cancer cells in patients via local elevation of temperature up to 85 °C [44,46]. High-intensity focused US is also an approved tool by the FDA for coagulative necrosis in the brain to create stereotactic lesions, also an irreversible ablation process [26].

We, along with some other research groups, have demonstrated that US stimulation on the PNS, with $I_{SPTA}$ between 1 and 200 W/cm$^2$, is unlikely to induce a sufficient temperature change in the target region to elicit temperature-driven neuromodulation [43,47]. This intermediate intensity range has been explored on the PNS by several neuromodulation studies, which have been systematically reviewed in the subsequent sections.

## 2.5. Metrics for Peripheral Neural Activities

The assessment of the neuromodulatory effects of US requires a reliable metric of peripheral neural activities, which includes direct and indirect measurements of neural action potentials from peripheral nerves or neurons. In addition, secondary effects of neural activities were also used to indirectly infer the neuromodulatory effects on PNS, including altered organ functions (i.e., bladder contraction, urethral sphincter relaxation), electroencephalogram recordings in the brain, and behavioral signs (e.g., whisker movement, freeze in motion, toe-pinch response). The detection of secondary effects tends to lag US stimulation by hundreds of milliseconds to seconds, a time frame much longer than the direct neuronal effects of milliseconds [38]. This review mostly focuses on the direct assessment of neural action potentials and omits metrics using secondary effects.

Nerve axons in the PNS are generally protected by soft connective tissue stacked in multiple layers, in contrast to the neurons and processes in the CNS, protected by bony structures like the skull or the vertebrae. These tightly wrapped tissue layers in the PNS, functioning as electrical insulators, pose a great challenge in recording electrophysiological activities from individual nerve axons, i.e., single-unit recordings [48]; recordings from individual neurons or axons in the CNS are straightforward when the electrodes are placed inside the skull or vertebrae. Consequently, the major metrics to assess the neuromodulatory effects on the PNS are either CAP, as a summation of action potentials from a bulk nerve bundle [47,49–52], or evoked muscle forces, as an indicator of motor nerve functions [42,53,54]. However, the characteristics of a CAP (peak amplitude, temporal location, and spread) depend on the temporal summation of a population of action potentials from axons with various spatial locations, morphologies, and insulation environment. The CAP characteristics can also be affected by changes in recording conditions, e.g., relative position of the electrode sites and axons, change of access impedance of recording electrodes due to altered moisture conditions, and multiple/chronic use of electrodes [48]. Thus, neither the changes in CAP amplitude nor the assumed changes in conduction delay (Figure 2A) can appropriately be representative of the effects of US neuromodulation. In addition, the signal strength in a CAP record can be misleading, as shown in Figure 2A: the large peak contributed to by fast-conducting A-fibers usually overshadows the small volleys by slow-conducting C-fibers, despite the presence of significantly higher proportions of C-fibers than A-fibers in the PNS [55,56]. Collectively, CAP appear to be an inappropriate metric for PNS neuromodulation. Further, muscle forces, evoked by US neuromodulation of a nerve, are indirect metrics of neve activities and limited to the study of motor axons innervating specific muscles.

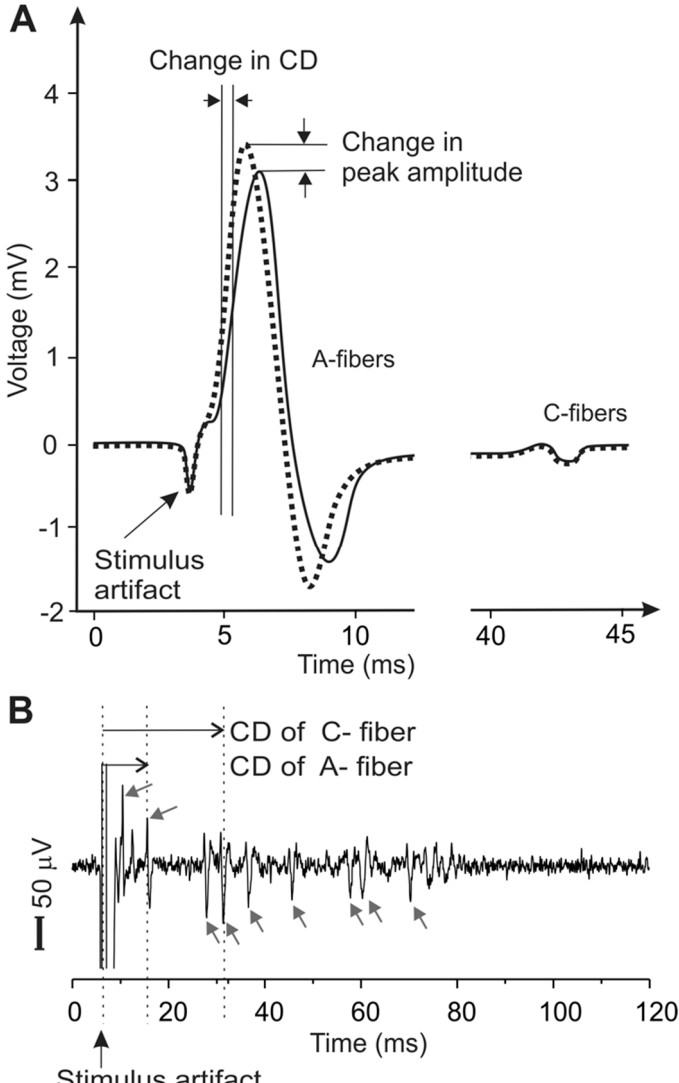

**Figure 2.** Metrics of peripheral neural activity via electrophysiological recordings of the compound action potentials (CAP) in (**A**) and single units in (**B**). CAP represent the temporal summation of multiple action potentials in the nerve trunk. Action potentials from individual nerve axons are recorded as distinct peaks in single-unit recordings (arrows), allowing a precise determination of the conduction velocities of individual axons for both myelinated A-fibers and unmyelinated C-fibers. CD: conduction delay.

On the other hand, single-unit recording can record action potentials from individual nerve axons (Figure 2B), capable of capturing relatively fine variations of neural responses to mechanical, chemical, and/or thermal stimuli [57–59]. Single-unit recording relies solely on the temporal information of the spike, which is mainly determined by the action potential transmission and is not affected by experimental artifacts like changes in electrode impedance. Thus, compared with CAP, single-unit recordings are more robust and provide much higher sensitivity to allow the detection of subtle changes of conduction delay in individual axons. However, single-unit recordings are technically challenging and involve microdissection of nerve fiber bundles until action potentials from a single nerve axon are isolated [43,48].

In general, the CAP recordings and evoked muscle activities are "macroscopic" detections of a large population of peripheral neural activities and thus may not serve as reliable and sensitive metrics for assessing the subtle changes of individual neural activities. The single-unit recordings possess a much higher sensitivity and are able to detect neuromodulatory effects within individual nerve axons,

i.e., at a "microscopic level". However, single-unit recordings are technically challenging and have only been implemented by a handful of studies [43,60–68], whereas CAP recordings have been widely used. Among the above single-unit studies, neuromodulation by ultrasound was assessed only by a recent study from us [43].

*2.6. Biological Preparations to Assess Peripheral US Neuromodulation*

Peripheral US neuromodulation has been studied both in humans and in preclinical animal models. Human studies benefit from the direct verbal feedback from the subjects, while preclinical animal studies allow mechanistic investigations in vivo, as well as in vitro or ex vivo with reduced and isolated systems. Studies that implement whole-animal in vivo preparations require to maintain anesthesia, the level of which can directly affect the detection of US neuromodulation [20]. In addition, the interpretation of the results from in vivo studies can be confounded by indirect effects of US on surrounding muscles, blood vessels, and immune cells. In vitro cultures of dissociated sensory neurons from the DRG have also been used as a model to study the effects of US neuromodulation [69]. Although the sensory cell somata in the DRG differ significantly from the nerve axons and endings in fundamental electrophysiological properties [59], neurites grown in cultured DRG have similar dimensions as axons and nerve endings and thus could potentially be an adequate model for studying peripheral neuromodulation. Nonetheless, cultured neurites in the absence of Schwann cells lack clustering of sodium channels [70] and thus differ significantly from the bundled axons in peripheral nerves in the physiological situation. In contrast, ex vivo studies on isolated peripheral nerves offer a physiologically relevant model for the direct assessment of US neuromodulatory effects, while avoiding the potential confounding factors of in vivo or in vitro studies. It is worth mentioning that many studies described isolated peripheral nerves as in vitro preparations, but they are considered as ex vivo preparations in this review to distinguish them from in vitro cultured DRG preparations.

## 3. Ultrasonic Neuromodulation of the Peripheral Nervous System

The effect of US stimulation to alter tissue activities was reported as early as in 1929 by a study on frogs and turtles [71]. The ability of low-frequency low-intensity US to modulate the CNS neural activities is elegantly demonstrated both in vivo in whole animals and in vitro in reduced systems of brain tissue slices (see reference [9] for a review). Recent studies indicate that US neuromodulation of the CNS might take an indirect route through the auditory cochlear pathway [29,30]. In stark contrast, the mechanisms of US neuromodulation of the PNS are still under debate, partly because of the contradictory experimental outcomes, which are systematically summarized and discussed below.

As summarized in Table 1, US appears to directly activate peripheral sensory nerve endings, as evidenced by studies of US stimulation of human hand, skin, soft tissues, bones, joints, ears, acupuncture points [53,72–75], as well as of cat ear [76] and frog Pacinian corpuscles [77]. CAP activities are evoked by US stimulation in animal preparations. More convincing evidence has been found by clinical studies in which direct verbal reports show that US stimulation is able to evoke virtually all the somatosensory modalities: tactile, warm, cold, itch, deqi, hearing, and pain. Unlike in the CNS, low-intensity US (<1 W/cm$^2$) is unable to activate mammalian nerve endings. In non-mammals, dissociated frog Pacinian corpuscles can be activated by US with intensity as low as 0.4 W/cm$^2$. US with intermediate intensity (1–200 W/cm$^2$) activates only low-threshold mechanoreceptors, e.g., tactile receptors and auditory nerve endings. Activation of other sensory modalities like temperature and pain generally requires high-intensity US stimulation (>1000 W/cm$^2$). However, it remains unclear whether US inhibits sensory nerve endings, which unlike neurons in the CNS, generally do not fire spontaneously. Further experimental studies are required to investigate whether US application to sensory nerve endings could lead to the loss of sensation.

**Table 1.** US neuromodulation of peripheral nerve endings. PNS: peripheral nervous system; PRF: pulse repetition frequency; DC: direct current.

| PNS Endings (Research) | Effect | Freq (MHz) | PRF (Hz) | DC (%) | Intensity (W/cm$^2$) | Duration (mSec) | Metric for Modulation | Preparation Type |
|---|---|---|---|---|---|---|---|---|
| Human hand nerve endings [72] (Gavrilov et al., 1977) | Tactile, warm, cold, itch, and pain sensation | 0.48 0.887 2.67 | | 100% | 160–30,000 | 1, 10, 100 | Verbal report | Clinical study |
| Human skin, soft tissue, bone, joint [73] (ab Ithel Davies et al., 1996) | Pain sensation | 0.48–2.67 | | | 12–15,000 | 1–100 | Verbal report | Clinical study |
| Human fingers and upper forearm [74] (Dalecki et al., 1995) | Tactile sensation | 2.2 | 50–1000 | 50% | 150 | 0.1 to 100 | Verbal report | Clinical study |
| Human ear [75] (Tsirulnikov et al., 1988) | Activation of acoustic nerve fibers | 2.5 | 125–8000 | 50% | 1–5 | 0.05–0.1 | Verbal report | Clinical study |
| Human acupuncture point [53] (Yoo et al., 2014) | Deqi sensation | 0.65 | 50 | 10% | 1–3 (SPPA) | 1000 | Verbal report | Clinical study |
| Cat ear [76] (Foster and Wiederhold, 1978) | Activation of auditory nerve | 5 | | 100% | 30 | 0.068 | CAP | In vivo |
| Frog Pacinian corpuscle [77] (Gavrilov et al., 1977) | Activation | 0.48 | | 100% | 0.4–2.5 | 0.1 to 100 | CAP | Ex vivo |

US neuromodulation of peripheral nerve trunks and axons has been investigated by a handful of studies summarized in Table 2, which give seemly contradictory results, with reports of enhanced nerve activities, suppressed activities, and mixed effects. Nonetheless, there are two consistent observations. First, all studies document no direct activation of peripheral nerves by US stimulation alone, except for a recent abstract report lacking technical details [78]. Second, high-intensity US stimulation (>200 W/cm$^2$) causes a nerve conduction block likely from a local thermal effect, and the nerve blocking effect can last for days to weeks and even be completely irreversible [50,54,79]. Lee et al. did report a reversible conduction block with a proper selection of the stimulus parameters [80]. The contradictory results occur with US intensity in the intermediate range, with enhanced nerve conduction velocity by some reports [43,51,81] and suppressed conduction velocity by others [47,52]; Mihran also reported that US enhanced CAP amplitude, thus providing mixed outcomes [52]. The difference in US neuromodulation effects could be attributed to the difference in the types of nerves studied (e.g., sciatic vs. vagus), preparations (e.g., ex vivo vs. in vivo), and animal species (e.g., mammals vs. non-mammals). CAP recordings are used to evaluate US effects except for one study in which single-unit recordings from individual axons are implemented [43]. As discussed earlier, single-unit recordings are more sensitive in detecting neuromodulatory effects than CAP recordings. Collectively, the study by Ilham et al. conducted on harvested nerve ex vivo with single-unit recordings has the least confounding factors and thus provides the most convincing results: US stimulation of intermediate intensity enhances peripheral nerve activity by increasing the conduction velocity in both A- and C-type axons [43]. This is further supported by a clinical study showing increased conduction velocity in human median nerves following US stimulation [81].

The DRG that contains sensory afferent somata has emerged as a promising target for neuromodulation [82–84]. To the best of our knowledge, all existing DRG neuromodulations implement electrical stimulation, and the modality of US has yet to be investigated on DRG. A recent pilot report with patch-clamp and calcium imaging recordings on dissociated DRG neurons shows that US evokes action potentials in 33–40% of DRG neurons, which may involve the activation of sodium, calcium, and non-selective ion channels [69].

**Table 2.** US neuromodulation of peripheral nerve axons.

| PNS Axons (Research) | Effect | Freq (MHz) | PRF (Hz) | DC (%) | Intensity (W/cm²) | Duration (mSec) | Metric for Modulation | Preparation Type |
|---|---|---|---|---|---|---|---|---|
| **Enhanced Activities** | | | | | | | | |
| Human median nerve [81] (Moore et al., 2000) | Increase conduction velocity | 1–3 | | 50–100% | 1 | 480,000 | Sensory and motor latency | Clinical study |
| Mouse sciatic nerve [43] (Ilham et al., 2018) | Increase conduction velocity | 1.1 | 200,000 | 20–40% | 0.91–28.2 | 40,000 | Single-unit | Ex vivo |
| Rat posterior tibia nerve [42] (Casella et al., 2017) | Inhibition of rhythmic bladder contraction | 0.25 | 2000 | | 0.9 (MPa) | 300 | Bladder contraction | In vivo |
| Bullfrog sciatic nerve [51] (Tsui et al., 2005) | Enhanced conduction, increased conduction velocity. | 3.5 | 2 | <1% | 1–3 W | | CAP | Ex vivo |
| Crab leg nerve [78] (Saffari et al., 2017) | Direct activation | | | | | | | Ex vivo |
| **Suppressed Activities** | | | | | | | | |
| Rabbit sciatic nerve [54] (Foley et al., 2007) | Nerve conduction block | 3.2 | | 58% | 1930 (SATA) | 10,000 | Flexion muscle force | In vivo |
| Rat sciatic nerve [79] (Foley et al., 2008) | Nerve conduction block | 5.7 | | | 390–7890 (SPTP) | 5000 | Muscle activities | In vivo |
| Rat sciatic nerve [80] (Lee et al., 2015) | Nerve conduction block | 2.68 | | | 2290–2810 (SATA) | 3000–7000 | CAP | Ex vivo |
| Rat vagus nerve [47] (Juan et al., 2014) | Inhibition of conduction, reduced conduction velocity | 1.1 | 20–1000 | | 18.7–93.4 | 15,000 | CAP | In vivo |
| Earth worm giant axon [85] (Wahab et al., 2012) | Inhibition of conduction, reduced conduction velocity | 0.825 | 100 | 10% | 0.1–0.7 (MPa) | 15,000–75,000 | CAP | In vivo |
| Frog sciatic nerve [86,87] (Young and Henneman, 1961) | Nerve conduction block | 2.7 | 0.33–0.5 | 11–30% | 1150 (SATA) | 9800 | CAP | Ex vivo |
| Bullfrog sciatic [50] (Colucci et al., 2009) | Nerve conduction block | 0.66–1.98 | 10, 20 | 1–20% | 370 | 30,000 | CAP | Ex vivo |
| **Mixed Effects** | | | | | | | | |
| Frog sciatic nerve [52] (Mihran et al., 1990) | Enhanced and suppressed excitability | 2–7 | 3–20 kHz | | 100–800 (SPTP) | 0.5 | CAP | Ex vivo |

## 4. Mechanisms of Peripheral US Neuromodulation

The mechanisms of action of US in the modulation of neural activities in the CNS have been systematically reviewed [9,38]. Here, we focus on US neuromodulation of peripheral nerve endings, axons, and somata in DRG, which, unlike the bone-protected CNS, are tightly wrapped by multiple layers of connective tissues, e.g., the epineurium, perineurium, endoneurium, and nerve extracellular matrix. This might explain why low-intensity US (<1 W/cm$^2$) directly activates brain neurons [9] but cannot activate peripheral nerve endings or axons. Another difference between the CNS and the PNS is the lack of inhibitory neurons in peripheral sensory afferents. In the CNS, both activation and suppression effects are reported by US stimulation [38], which can be attributed to the selective activation of excitatory (e.g., glutamatergic) and inhibitory (e.g., GABAergic and glycinergic) neurons, respectively. In the periphery, the suppression effect is generally reflected as inhibition of action potential generation or transmission [50].

Sensory nerve endings can be directly activated by US stimulation, as evidenced by a series of classical clinical studies by Gavrilov et al., which reveal that virtually all the sensory modalities can be activated by US at intermediate and high-intensity [72,73]. The activation of the low-threshold mechanoreceptors in the skin and cochlear hair cell appears to require the least US intensity and energy [72,75,76], indicating that US likely evokes action potentials via mechanotransduction intrinsic to those nerve endings, i.e., opening of mechanosensitive ion channels by the local mechanical force leading to action potential generation in the spike initiation zone [59]. Evoking other sensory modalities requires high-intensity focused US, which can lead to significant local heating (>5 °C) and inertial cavitation with sudden collapse of the bubble [88]. How high-intensity US activates sensory nerve endings remains undetermined. Putative mechanisms include temperature gating of voltage-sensitive sodium and potassium ion channels, mechanical gating of other transducer molecules like the transient receptor potential (TRP) channels, and indirect effects on surrounding non-neuronal tissues.

US neuromodulation to block action potential transmission in peripheral nerve axons has been extensively studied and appears to require the local thermal effect of high-intensity US stimulation [49,50]. The rise of local temperature reportedly induces a conduction block in peripheral axons by changing the kinetics of voltage-sensitive sodium channels, leading to their inactivation; temperature has a much greater impact on the inactivation kinetic than on the activation kinetic [89]. Non-thermal mechanisms could also contribute to the peripheral nerve block by high-intensity US, e.g., inertia cavitation with strong acoustic forces that directly "bombs" the fibers leading to irreversible disruption [49]. Despite the invasiveness, clinical applications of high-intensity US neuromodulation on peripheral nerves show beneficial effects on pain management in painful amputation stump neuromas, phantom limbs [90], and spasticity [91].

Reversible peripheral neuromodulation of peripheral nerve axons implements low- and intermediate-intensity US which does not directly evoke action potentials [43,47]. Intermediate-intensity US generally causes a negligible thermal effect (<1 °C) and produces harmless stable cavitation [43,47], and thus, its neuromodulatory effects on peripheral nerve axons occur likely through local acoustic radiation forces. From analyzing the existing experimental reports in Table 2, intermediate-intensity US likely enhances the neural activity in mammalian peripheral nerves by increasing the nerve conduction velocity. Unlike the nerve endings, the axon may lack the mechanosensitive ion channels tuned to transduce micromechanical forces. Thus, it is likely that other mechanically gated ion channels may participate to collectively enhance the neural activity, which include, but are not limited to, voltage-sensitive sodium [92–94], two-pore-domain potassium (K2P) [95], ASIC [95], TRP [96], and Piezo channels [97,98].

Virtually no study has been conducted to assess US neuromodulation on intact DRG, the clustering of sensory afferent somata. A recent study indicates that action potentials can be evoked by focused US in dissociated DRG neurons [69], opening new avenues of research on US DRG neuromodulation in future preclinical and clinical studies. The underlying mechanisms of US activation of DRG neurons are unclear and likely involve sodium, potassium, and non-selective cation channels [69].

The non-thermal and non-cavitation bio-effect of the focused US at low and intermediate intensity is of central interests for reversible US neuromodulation; it likely induces local acoustic forces below the harmful range. Potential mechanisms of neuromodulation include the mechanical gating of transmembrane ion channels, as discussed earlier. In addition, several theories have attributed the US neuromodulatory effect to altered properties of the lipid-bilayer membrane at the nerve endings, axons, and somata, including the soliton model [99], the flexoelectricity hypothesis [100], the neuronal intramembrane cavitation excitation (NICE) model [101], and, more recently, the theory of direct transmembrane pore formation [102]. The main assumption of the soliton model is that the transmission of signal through a nerve occurs as an electromechanical soliton wave packet rather than a complete electrical phenomenon. However, this model could not mathematically explain the role of voltage-gated ion channels in action potential generation. The flexoelectric effect hypothesizes that the mechanical energy of the curved lipid-bilayer membrane leads to electrical membrane polarization to depolarize the neural membrane, but the relevant mathematical formulation to account for action potential generation has yet to be established. In the NICE model, it is hypothesized that US stimulation with sufficient intensity ($>0.10$ W/cm$^2$ $I_{SPTA}$) causes nanobubble formation in the intramembrane space, which subsequently changes the transmembrane capacitance. Hence, the NICE model suggests the membrane capacitive current caused by the change of transmembrane capacitance as the source of US neuromodulation and appears to explain the increased conduction velocities by US modulation from our recent study with single-unit recordings [43]. The recent experimental finding on an expression system indicates that US as low as 0.4 W/cm$^2$ can form pores in the lipid-bilayer membrane large enough to allow the passage of the large dye molecule calcein [102], indicating that the pores are sufficiently large for the passage of sodium and potassium ions to excite neurons. Both the NICE model and the transmembrane pore formation theory can explain the enhanced excitability of peripheral nerve axons by US, consistent with a recent ex vivo study with single-unit recordings from individual axons [43]. Further experimental and theoretical studies are required to advance our mechanistic understanding of peripheral US neuromodulation.

## 5. Conclusions

Peripheral US neuromodulation is capable of both enhancing and suppressing neural activities which are likely dependent upon the range of US intensity and strength. Unlike the neurons in the brain, low-intensity US ($<1$ W/cm$^2$) is unable to evoke action potentials in the peripheral nerve endings or axons. US of intermediate intensity (1 to 200 W/cm$^2$) exerts mainly acoustic radiation force on tissues, with no apparent thermal or inertia cavitation effects. US of intermediate intensity activates low-threshold mechanosensitive nerve endings likely through the regular mechanotransduction process, by opening mechanosensitive ion channels to evoke action potentials. US of intermediate intensity also enhances the neural activity of peripheral nerve axons, leading to increased nerve conduction velocities in both A- and C-type fibers, which is likely caused by mechanical gating of other ion channels like the NaV, K2P, ASIC, TRP, and Piezo channels. In addition, enhanced neural activity could be attributed to a direct effect of acoustic radiation force on the lipid-bilayer neural membrane. Plausible mechanisms include a transient capacitive current from rapid changes of local membrane capacitance (the NICE model) and transmembrane pore formation to allow sodium and potassium ions to pass through. High-intensity US ($>1000$ W/cm$^2$) consistently inhibits the action potential transmission in peripheral nerves (i.e., nerve block) likely through a thermal effect. In addition, inertia cavitation from high-intensity US could lead to irreversible damage of peripheral nerve axons. In conclusion, US neuromodulation of the PNS has profound therapeutic potential especially through the non-thermal non-cavitation bio-effect in the intermediate intensity range, which is able to non-invasively and reversibly enhance peripheral neural activities.

**Author Contributions:** Conceptualization, B.F., L.C., and S.J.I.; investigation, B.F., L.C., and S.J.I.; writing—original draft preparation, B.F.; writing—review and editing, B.F., L.C., and S.J.I.; visualization, B.F. and S.J.I.; supervision, B.F.; funding acquisition, B.F.

**Funding:** This research was funded by the U.S. National Institutes of Health, grant number K01 DK100460 and R03 DK114546; U.S. National Science Foundation, grant number 1727185.

**Conflicts of Interest:** The authors declare no conflict of interest.

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
