# Peer review of "A Review on Ultrasonic Neuromodulation of the Peripheral Nervous System: Enhanced or Suppressed Activities?"

_applsci, doi:10.3390/app9081637_

Round 1

Reviewer 1 Report

The previous comments are well addressed. Fig 2 needs reformatting.

Author Response

We thank the reviewer for agreeing on the revisions we have made on the manuscript.

Reviewer 2 Report

General Comments

The authors provide a generally comprehensive review of numerous studies in the field of ultrasonic neuromodulation. Although the review focus on the peripheral nervous system, the suggestion that the mode of action for peripheral and central nervous system may be delivered by different mode of action is interesting. Therefore I would strongly suggest the authors to slightly develop this point and add a few of reference for the review and the coverage of the more recent ultrasonic neuromodulation literature (in particular in animals other than rodents):

1) Neuromodulation Literature & Discussion
Though the authors cover neuromodulation effect of ultrasound in the introduction, I was surprised that researches combining extracellular electrophysiological recordings in awake monkey was not discussed (Wattiez et al. 2017) as well as functional MRI (Verhagen et al. 2019). This is an important point to discuss especially in references to ultrasonic parameters and neurons/fibers mode of action.

2) Utility of Figures
The Figure is rather superficial and not helpful in guiding the reader through the reviewed literature. Figure 2 is at it is unreadable (black background) and I would strongly recommend to add additional figures in part 3.

Author Response

We thank the reviewer for the constructive comments and valid concerns about the manuscript. We have revised point-by-point to the reviewer’s questions as follows. The manuscript has been modified accordingly with changes tracked.

General Comments

The authors provide a generally comprehensive review of numerous studies in the field of ultrasonic neuromodulation. Although the review focus on the peripheral nervous system, the suggestion that the mode of action for peripheral and central nervous system may be delivered by different mode of action is interesting. Therefore I would strongly suggest the authors to slightly develop this point and add a few of reference for the review and the coverage of the more recent ultrasonic neuromodulation literature (in particular in animals other than rodents):

We really appreciate the reviewer’s positive comments on the study! We have included the two citations in the revision on ultrasonic neuromodulation of macaque as suggested by the reviewer.

1) Neuromodulation Literature & Discussion
Though the authors cover neuromodulation effect of ultrasound in the introduction, I was surprised that researches combining extracellular electrophysiological recordings in awake monkey was not discussed (Wattiez et al. 2017) as well as functional MRI (Verhagen et al. 2019). This is an important point to discuss especially in references to ultrasonic parameters and neurons/fibers mode of action.

We thank the reviewer for drawing our attention to these two important studies. We have added the following sentences to the introduction at line 60 in the revision.

“In addition, transcranial ultrasonic stimulation (TUS) as a non-invasive brain stimulation technique has been investigated in primates. TUS of the frontal eye field evokes transient increase of neural activities in the supplementary eye field in awake macaques as measured by multi-channel single-unit recordings (1). In addition, a 40-second-long TUS of certain brain regions can lead to long-lasting neuromodulatory effect up to two hours post stimulation according to a functional MRI study in macaques (2). Especially, TUS of the supplementary motor area and frontal polar cortex caused those brain regions to interact more selectively with the rest of the brain.”

2) Utility of Figures
The Figure is rather superficial and not helpful in guiding the reader through the reviewed literature. Figure 2 is at it is unreadable (black background) and I would strongly recommend to add additional figures in part 3.

We have went back to check the submitted Fig. 2 and found it quite readable with white background. Perhaps Fig. 2 was not delivered to the reviewer properly. To clarify, we have included Fig. 2 in a high-resolution BMP format in the revision.

References

1.            Wattiez N, Constans C, Deffieux T, Daye PM, Tanter M, Aubry JF, et al. Transcranial ultrasonic stimulation modulates single-neuron discharge in macaques performing an antisaccade task. Brain stimulation. 2017 Nov - Dec;10(6):1024-31. PubMed PMID: 28789857. Epub 2017/08/10. eng.

2.            Verhagen L, Gallea C, Folloni D, Constans C, Jensen DE, Ahnine H, et al. Offline impact of transcranial focused ultrasound on cortical activation in primates. eLife. 2019 Feb 12;8. PubMed PMID: 30747105. Pubmed Central PMCID: PMC6372282. Epub 2019/02/13. eng.

This manuscript is a resubmission of an earlier submission. The following is a list of the peer review reports and author responses from that submission.

Round 1

Reviewer 1 Report

The authors addressed „Ultrasonic neuromodulation of the peripheral nervous system:

enhanced or suppressed activities?“ and by submitting the manuscript to “Applied Sciences” they chose a very well suited journal for this topic.

The authors performed a detailed literature search and wrote an easily understandable text. The topic is in detail reviewed – with a very well understandable excurse to neuromodulation-techniques in general - and I did not miss any aspect regarding US affecting PNS.

In the comments below I suggest modifications and highlighted spelling/wording/gramma mistakes in bold.

Page 2 line 53:”…a relatively low frequency…”: one can discuss if Mhz is low J; suggestion: “…ultrasound (US) as a mechanical wave operating at 250 KHz to 50 MHz allows spatially…”

Page 2 line 83 to page 3 line 95: maybe this is a copy/paste mistake. Please delete this material and methods section.

Page 3 line 96: neuroulation. Please modify to neuromodulation

Page 6 line 219: please check punctuation

Page 6 line 222: change “is” to “are”

Section 2.5: please add a comment on the relevance of neuromodulation detected using single unit recording versus CAP. One may speculate that the recording of one fiber is irrelevant for subsequent activation of higher order structures in contrast to detection of synchronized activation of a nerve bundle which is often needed for central activation. Summary: single unit gives more details but CAP is more relevant for neural activity.

Page 7 line 239: please change “has” to “have”

Page7 line 242: here you stated that DRG is not a feasible model since the somata react different from “the nerve axons and endings”. This is true but not relevant since you should compare the DRG neurites with your axons and dendrites. DRG may be a good model if you investigate the relevant structures.

Page 7 line 247: please add “system”

Page 7 line 285ff: not understandable. Please rewrite the sentence.

Author Response

We thank the reviewer for the constructive comments and valid concerns about the manuscript. We have revised point-by-point to the reviewers’ questions as follows. The manuscript has been modified accordingly with changes tracked.

Review1

The authors addressed “Ultrasonic neuromodulation of the peripheral nervous system: enhanced or suppressed activities?” and by submitting the manuscript to “Applied Sciences” they chose a very well suited journal for this topic.

The authors performed a detailed literature search and wrote an easily understandable text. The topic is in detail reviewed – with a very well understandable excurse to neuromodulation-techniques in general - and I did not miss any aspect regarding US affecting PNS.

 We really appreciate the reviewer’s positive comments on the study!

In the comments below I suggest modifications and highlighted spelling/wording/gramma mistakes in bold.

 We thank the reviewer for pointing these mistakes out and have modified accordingly.

Page 2 line 53:”…a relatively low frequency…”: one can discuss if Mhz is low; suggestion: “…ultrasound (US) as a mechanical wave operating at 250 KHz to 50 MHz allows spatially…”

 Following the reviewer’s suggestion, the corresponding sentence has been edited as “On the other hand, ultrasound (US) as a mechanical wave operating at 250 KHz to 50 MHz allows spatially and temporally precise delivery of energy deep into the tissue. Hence, US has been considered as an ideal means for noninvasive neuromodulation.”.

Page 2 line 83 to page 3 line 95: maybe this is a copy/paste mistake. Please delete this material and methods section.

We regret leaving these paragraphs when we formatted the manuscript using the journal’s template. We have deleted those paragraphs in the revision.

Page 3 line 96: neuroulation. Please modify to neuromodulation

This typo has been corrected as “neuromodulation”.

Page 6 line 219: please check punctuation

We have rewritten the sentences as follows:

“Single-unit recording relies solely on the temporal information of the spike, which is mainly determined by the action potential transmission and not affected by experimental artifacts like changes in electrode impedance. Thus, compared with CAP, single-unit recordings are more robust and provide much higher sensitivity to allow the detection of subtle changes of conduction delay in individual axons.”

Page 6 line 222: change “is” to “are”

We have replaced “is” with “are” in the corresponding sentence.

Section 2.5: please add a comment on the relevance of neuromodulation detected using single unit recording versus CAP. One may speculate that the recording of one fiber is irrelevant for subsequent activation of higher order structures in contrast to detection of synchronized activation of a nerve bundle which is often needed for central activation. Summary: single unit gives more details but CAP is more relevant for neural activity.

We thank the reviewer for raising this point. We have rewritten the paragraph as follows to emphasize that single-unit recordings allow detailed microscopic detection of neuromodulation whereas CAP is a metric for macroscopic neural activities.

“In general, the CAP recordings and evoked muscle activities are “macroscopic” detections of a large population of peripheral neural activities, and thus may not serve as reliable and sensitive metrics for assessing the subtle changes of individual neural activities. The single-unit recordings possess a much higher sensitivity able to detect neuromodulatory effects within individual nerve axons, i.e., at “microscopic level”. However, single-unit recordings are technically challenging and have only been implemented by a handful of studies (40, 55-63), whereas CAP recordings were widely used.”

Page 7 line 239: please change “has” to “have”

We have corrected “has” as “have” in the corresponding sentence.

Page7 line 242: here you stated that DRG is not a feasible model since the somata react different from “the nerve axons and endings”. This is true but not relevant since you should compare the DRG neurites with your axons and dendrites. DRG may be a good model if you investigate the relevant structures.

We agree with the reviewer that cultured neurites from DRG could be an adequate model for studying peripheral neuromodulation and have modified the sentences as follows:

“Although the sensory cell somata in the DRG differ significantly from the nerve axons and endings in fundamental electrophysiological properties (54), the neurite outgrowth in cultured DRG has similar dimensions as axons and nerve endings and thus could potentially be an adequate model for studying peripheral neuromodulation. Nonetheless, cultured neurites in the absence of Schwann cells lack clustering of sodium channels (65) and thus differ significantly from the bundled axons in peripheral nerves in the physiological situation.”

Page 7 line 247: please add “system”

We have added system in the corresponding location.

Page 7 line 285ff: not understandable. Please rewrite the sentence.

We have modified the sentence as “As discussed earlier, single-unit recordings are more sensitive in detecting neuromodulatory effects than the CAP recordings.”.

Reviewer 2 Report

This manuscript describes some issues related to ultrasonic stimulation for neuromodulation.

This area of research is growing rapidly and could become very important for a variety of applications. This paper summarizes some aspects of this area and could make a generally useful addition to the literature.

However, I have two principle concerns.  One is that it is not clear what, if any, new information is provided in this manuscript.  It all looks familiar to some degree.  What is really new?

The second, and more concerning, is the really poor job of editing done by the authors before submission.  I have reviewed and edited many manuscripts, and this is by far the worse, and  laughably so.  This paragraph was actually found in the manuscript, starting at Line 84:

"Materials and Methods should be described with sufficient details to allow others to replicate and build on published results. Please note that publication of your manuscript implicates that you must make all materials, data, computer code, and protocols associated with the publication available to readers. Please disclose at the submission stage any restrictions on the availability of materials or information. New methods and protocols should be described in detail while well-established methods can be briefly described and appropriately cited. "

There are a variety of other glaring problems.  If the authors missed these gross errors, what more subtle details are missing or also in error?

Other issues:

"The NICE model appears to be the most convincing one, "  There is little data given to support this claim.

The authors suggest that ultrasound is completely safe, but there are concerns about heating, disruption of cell membranes, etc. that should be mentioned.

Author Response

We thank the reviewer for the constructive comments and valid concerns about the manuscript. We have revised point-by-point to the reviewers’ questions as follows. The manuscript has been modified accordingly with changes tracked.

Review 2

This manuscript describes some issues related to ultrasonic stimulation for neuromodulation.

This area of research is growing rapidly and could become very important for a variety of applications. This paper summarizes some aspects of this area and could make a generally useful addition to the literature.

 We thank the reviewer for considering this study as a useful addition to the literature.

However, I have two principle concerns.  One is that it is not clear what, if any, new information is provided in this manuscript.  It all looks familiar to some degree.  What is really new?

We agree with the reviewer that the US neuromodulation has been systematically reviewed and studied especially with effects on the central nervous system. But it remains a gap in our knowledge regarding the neuromodulatory effects of US on peripheral nerve endings, axons and somata in the DRG. Especially, there are controversial reports in the literature with some showing enhanced peripheral nerve activities and others reporting suppressed activities. This review follows our recent study (Ilham et al., 2018) that used single-unit recordings to definitively reveal the role of US to enhance the conduction velocities of both A- and C- type peripheral axons in the absence of appreciable thermal effects. This review also draws attention to the limitations of compound action potential (CAP), the major detection method used previously to assess peripheral neuromodulation. CAP is not solely determined by action potential spikes but can be profoundly affected by recording artifacts like the impedance of the electrodes. To the best of our knowledge, no other reviews of US neuromodulation have paid attention to the different detection methods and discussed their potential influences on the experimental outcomes.

      The second, and more concerning, is the really poor job of editing done by the authors before submission.  I have reviewed and edited many manuscripts, and this is by far the worse, and  laughably so.  This paragraph was actually found in the manuscript, starting at Line 84: 

"Materials and Methods should be described with sufficient details to allow others to replicate and build on published results. Please note that publication of your manuscript implicates that you must make all materials, data, computer code, and protocols associated with the publication available to readers. Please disclose at the submission stage any restrictions on the availability of materials or information. New methods and protocols should be described in detail while well-established methods can be briefly described and appropriately cited. "

There are a variety of other glaring problems.  If the authors missed these gross errors, what more subtle details are missing or also in error?

We apologize for this mistake. The manuscript was prepared in a different format. Trying to meet the deadline, we were pressed by time to reformat using the template for the journal of “Applied Sciences” and forgot to delete those paragraphs from the template. We have done so in this revision.

Other issues:

"The NICE model appears to be the most convincing one, "  There is little data given to support this claim.

Following the reviewer’s suggestion, we have modified the sentence as:

“The NICE model suggests the membrane capacitive current caused by the change of transmembrane capacitance as the source of US neuromodulation, and appears to explain the increased conduction velocities by US modulation from our recent study with single-unit recordings (40).”

The authors suggest that ultrasound is completely safe, but there are concerns about heating, disruption of cell membranes, etc. that should be mentioned.

We thank the reviewer for raising these points. We have not claimed that ultrasound is completely safe, we suggest to use intermediate intensity to modulate nerve activities, “US of intermediate intensity (1 to 200 W/cm2) exerts mainly acoustic radiation force on tissues with no apparent thermal or inertia cavitation effects.” and also “In conclusion, the US neuromodulation of the PNS has profound therapeutic potential especially for the non-thermal non-cavitation bio-effect in the intermediate intensity range, which is capable to non-invasively and reversibly enhance the peripheral neural activities.”. In the review, we also fully acknowledge the deleterious effects of heating and disruption of cell membranes by high-intensity US: “High-intensity US (>1000 W/cm2) consistently inhibits the action potential transmission in peripheral nerves (i.e., nerve block) likely due to a thermal effect. In addition, inertia cavitation from high-intensity US could lead to irreversible damage of peripheral nerve axons.”
